# Efficient Reward Poisoning Attacks on Online Deep Reinforcement Learning

**Yinglun Xu**                                                      *yinglun6@illinois.edu*
*Department of Computer Science*
*University of Illinois at Urbana-Champaign*

**Qi Zeng**                                                          *qizeng2@illinois.edu*
*Department of Computer Science*
*University of Illinois at Urbana-Champaign*

**Gagandeep Singh**                                                 *ggnds@illinois.edu*
*Department of Computer Science*
*University of Illinois at Urbana-Champaign*

**Reviewed on OpenReview:** *https://openreview.net/forum?id=25G63lDHV2*

## Abstract

We study reward poisoning attacks on online deep reinforcement learning (DRL), where the attacker is oblivious to the learning algorithm used by the agent and the dynamics of the environment. We demonstrate the intrinsic vulnerability of state-of-the-art DRL algorithms by designing a general, black-box reward poisoning framework called adversarial MDP attacks. We instantiate our framework to construct two new attacks which only corrupt the rewards for a small fraction of the total training timesteps and make the agent learn a low-performing policy. We provide a theoretical analysis of the efficiency of our attack and perform an extensive empirical evaluation. Our results show that our attacks efficiently poison agents learning in several popular classical control and MuJoCo environments with a variety of state-of-the-art DRL algorithms, such as DQN, PPO, SAC, etc.

## 1 Introduction

To obtain state-of-the-art performance in many real-world environments such as robot control (Christiano et al., 2017) and recommendation systems (Afsar et al., 2021; Zheng et al., 2018), online deep reinforcement learning (DRL) algorithms relying on human feedback for determining rewards have been applied. The dependency on human feedback introduces the threat of reward-based data poisoning during training: a user can deliberately provide malicious rewards to make the DRL agent learn low-performing policies. Data poisoning has already been identified as the most critical security concern when employing learned models in industry (Kumar et al., 2020). Thus, it is essential to study whether state-of-the-art DRL algorithms are vulnerable to reward poisoning attacks to discover potential security vulnerabilities and motivate the development of robust training algorithms.

**Black-box DRL attack with limited budgets.** To uncover practical vulnerabilities, it is critical that the attack does not rely on unrealistic assumptions about the attacker's capabilities. Therefore to ensure a practically feasible attack, we require that: (i) the attacker does not know the exact algorithm used by the agent and the parameters of the neural network used for training. The attack should work for different kinds of learning algorithms (e.g., policy optimization, Q learning), (ii) the attacker does not know the dynamics of the agent's environment, (iii) to ensure stealthiness, the amount of reward corruption applied by the attacker over each step, each episode, and the whole training process should be limited (see Section 3), and (iv) the attacker does not have access to significant memory or computing resources to adjust strategy for reward perturbations during training.

**Challenges in poisoning attack against DRL agents.** The biggest challenge in the DRL setting compared to traditional RL settings such as bandits and tabular MDPs comes from the complexity of the DRL environment and algorithms. In attack formulations for traditional RL settings, the complexity of the attack problem and the optimality-gap of the solution scale with the number of states and actions of the environment (Rakhsha et al., 2020; Zhang et al., 2020b; Banihashem et al., 2022). These numbers become infinite in the DRL setting where the state and action spaces are continuous and high dimensional. The aforementioned problems become too complicated to solve, and no guarantee of optimality can be given for the solutions. Furthermore, the computational resource required by previous attacks for traditional RL settings can be as or even more than that required by the learning agents themselves (Zhang et al., 2020b; Rakhsha et al., 2021), which is not practical for Deep RL applications. Finally, the learning algorithms in the traditional RL settings are well-understood, making their vulnerabilities more explicit for the attacker to exploit. In the DRL settings, the theoretical understanding of the state-of-the-art DRL algorithms is still limited.

**This work: efficient poisoning attacks on DRL.** To the best of our knowledge, no prior work studies the vulnerability of the DRL algorithms to reward poisoning attacks under the practical restrictions mentioned above. To overcome the challenges in designing efficient attacks and demonstrate the vulnerability of the state-of-the-art DRL algorithms, we make the following contributions:

1. We propose a general, efficient, and parametric reward poisoning framework for the general RL setting, which we call adversarial MDP attack, and instantiate it to generate two attack methods that apply to any kind of learning algorithms and are computationally efficient. To the best of our knowledge, our attack is the first to consider the following five key elements in the threat model at the same time: 1. Training time attack, 2. Online deep RL, 3. Reward poisoning attack, 4. Complete black box attack (no knowledge about the learning algorithm and the dynamics of the environment), and 5. Limited computational budget.

2. We provide a theoretical analysis for the efficiency of our attack methods in the general RL setting under basic assumptions on the efficiency of the algorithms.

3. We provide an extensive evaluation of our attack methods for poisoning the training process of several state-of-the-art DRL algorithms, such as DQN, PPO, SAC, etc., learning in classical control and MuJoCo environments that are commonly used for developing and testing DRL algorithms. Our results show that our attack methods significantly reduce the performance of the policy learned by the agent in most cases.

## 2 Related Work

**Testing time attack on RL.** Testing time attacks (evasion attack) in the deep RL setting is popular in literature (Huang et al., 2017; Kos & Song, 2017; Lin et al., 2017). For an already trained policy, testing time attacks find adversarial examples where the learned policy has undesired behavior. In contrast, our training time attack corrupts reward to make the agent learn low-performing policies.

**Data poisoning attack on bandit and tabular RL settings.** Data poisoning attacks have been studied in the simpler bandit Jun et al. (2018); Liu & Shroff (2019); Xu et al. (2021b) and tabular RL Ma et al. (2019); Rakhsha et al. (2020); Zhang et al. (2020b); Liu & Lai (2021); Xu et al. (2021a) settings. These attacks require the state and action spaces of the environment to be finite, making them not applicable to the deep RL setting.

**Observation perturbation attack and defense.** There is a line of work studying observation perturbation attacks during training time (Behzadan & Munir, 2017a;b; Inkawhich et al., 2019) and its corresponding defense (Zhang et al., 2021a; 2020a). The threat model here does not change the actual state or reward of the environment, but instead, it changes the learner's observation of the environment by generating adversarial examples. In contrast, for the poisoning attack as considered in our work, the attacker changes the actual reward or state of the environment. Also, the observation perturbation attack mainly utilizes the vulnerability of the neural network to adversarial examples, which is only possible through state poisoning attacks.

In our work, we focus on another vulnerability that is more related to the attack considered in reinforcement learning previously. This vulnerability is that an algorithm may not sample the environments enough under the attack and conclude a sub-optimal policy to be the optimal one by the end of training. We exploit this vulnerability through reward poisoning attacks, and in principle, both state and reward poisoning attacks can utilize such vulnerability.

**Data poisoning attack on DRL.** The work of Sun et al. (2020) is the only other work that considers reward poisoning attack on DRL and therefore is the closest to ours. First of all, the attack problem is different in terms of budget. In Sun et al. (2020), the attacker tries to minimize the perturbation per training batch while in our work, the attacker tries to minimize the perturbation per step and over the whole training process. In addition, there are four main limitations of their attack compared to ours (a) the attack requires the knowledge of the learning algorithm (the update rule for learned policies) used by the agent, which is not available in the black box setting, (b) the attack only works for on-policy learning algorithms, (c) the computational resource required by their attacker is more than that required by the learning agent and (d) the attacker receives the whole training batch before deciding perturbations. This makes the attack infeasible when the agent updates the observation at each time step, as it is impossible for the attacker to apply corruption to previous observations in a training batch. Since there is no other work we could find for our more realistic attack settings, we artificially adapt our attack to the setting in their work in order to experimentally compare our results to theirs. Our results in Appendix D show that our attack requires less computational resources and achieves better attack results.

**Robust learning algorithms against data poisoning attack.** Robust learning algorithms can guarantee efficient learning under the data poisoning attack. There have been studies on robustness in the bandit (Lykouris et al., 2018; Gupta et al., 2019), and tabular MDP settings (Chen et al., 2021; Wu et al., 2021; Lykouris et al., 2021), but these results are not applicable in the more complex DRL setting. For the DRL setting, Zhang et al. (2021b) proposes a learning algorithm guaranteed to be robust in a simplified DRL setting under strong assumptions on the environment (e.g., linear Q function and finite action space). The algorithm is further empirically tested in actual DRL settings, but the attack method used for testing robustness, which we call reward flipping attack, is not very efficient and malicious as we show in Appendix C. Testing against weak attack methods can provide a false sense of security. Our work provides attack methods that are more suitable for empirically measuring the robustness of learning algorithms.

## 3   Background

**Reinforcement learning.** We consider a standard RL setting where an agent trains by interacting with an environment Sutton & Barto (2018). The interaction involves the agent observing a state representation of the environment, taking an action, and receiving a reward. Formally, an environment is represented by a stationary Markov decision process (MDP), $\mathcal{M} = \{\mathcal{S}, \mathcal{A}, \mathcal{P}, \mathcal{R}, \mu\}$, where $\mathcal{S}$ is the state space, $\mathcal{A}$ is the action space, $\mathcal{P} : \mathcal{S} \times \mathcal{A} \to D(\mathcal{S})$ is the state transition function, $\mathcal{R} : \mathcal{S} \times \mathcal{A} \to D(\mathbb{R})$ is the reward function, and $\mu$ is the distribution of the initial states. The training process consists of multiple episodes where each episode is initialized with a state sampled from $\mu$, and the agent interacts with the environment in each episode until it terminates. A policy $\pi : \mathcal{S} \to D(\mathcal{A})$ is a mapping from the state space to the space of probability distribution $D(\mathcal{A})$ over the action space. If a policy $\pi$ is deterministic, then $\pi(s)$ returns a single action. A value function $V_{\mathcal{M}}^{\pi}(s)$ is the expected reward an agent obtains by following the policy $\pi$ starting at state $s$ in the environment $\mathcal{M}$. We denote $\mathcal{V}_{\mathcal{M}}^{\pi} := \mathbb{E}_{s_0 \sim \mu} V_{\mathcal{M}}^{\pi}(s_0)$ as the policy value for a policy $\pi$ in $\mathcal{M}$, which measures the performance of $\pi$. Let $\Pi$ be the set of all possible policies, the goal of the RL agent is to find the optimal policy with the highest policy value $\pi^* = \arg\max_{\pi \in \Pi} \mathcal{V}_{\mathcal{M}}^{\pi}$. Correspondingly, we denote $V^* = \mathcal{V}_{\mathcal{M}}^{\pi^*}$ as the performance of the optimal policy. We say a policy $\pi$ is $\epsilon$-optimal if its policy value satisfies $\mathcal{V}_{\mathcal{M}}^{\pi} \geq V^* - \epsilon$.

**Reward poisoning attack on RL.** In this work, we consider a standard data poisoning attack setting (Jun et al., 2018; Rakhsha et al., 2020) where a malicious adversary tries to manipulate the agent by poisoning the reward received by the agent from the environment during training. Formally, at round $t$, let state and action taken by the agent be $s^t$ and $a^t$. The environment generates the instant reward $r^t \sim \mathcal{R}(s^t, a^t)$ and next state $s^{t+1} \sim \mathcal{S}(s^t, a^t)$. The attacker injects perturbation $\Delta^t$ to the reward. Then the agent receives the

perturbed observation $(s^t, a^t, r^t + \Delta^t, s^{t+1})$. Next, we describe the constraints on the attacker's knowledge, attack budgets, and the attack goal.

**Constraints on attacker**: We consider a fully black-box attack setting with limitations on computation resources as follows.

1. **Oblivious to the algorithm.** The attacker has no knowledge of the training algorithm including any parameters in the network used by the agent while training.

2. **Oblivious to the environment.** The attacker has no knowledge about the MDP $\mathcal{M}$ except for the state and actions spaces $\mathcal{S}$ and $\mathcal{A}$.

3. **Limited computational resources.** The attacker can observe the current state, action, and reward tuple $(s^t, a^t, r^t)$ generated during training at each timestep $t$. However, it does not have significant memory or computing resources to store and learn based on the information from historical observations.

We believe such constraints make an attack realistic. In practice, the learning algorithm used by an agent is often private. Furthermore, in many cases, the dynamics of the environment can be unknown even to the learning agent, not to mention the attacker. Finally, the attacker may not always have access to significant computational resources for constructing the attack.

**Attacker's budget**: We consider two budgets for the attack that are typical and have been considered in previous works (Zhang et al., 2020b; Rakhsha et al., 2020). The attacker tries to minimize the budgets during the attack.

1. **Number of corrupted training steps.** The number of timesteps $C = \sum_{t=1}^{t} \mathbb{1}\{|\Delta^t| > 0\}$ corrupted by the attack.

2. **Bounded per-step corruption.** The maximum corruption at each timestep $B = \max_t |\Delta^t|$.

We notice that it can be even more realistic if one also considers the total corruption across an episode as part of the budget. To simplify the attack model for theoretical analysis, we do not consider the per-episode perturbation budget in Section 4, 5, and we consider it for the experiments in Section 6.

**Attacker's goal**: Let $\pi_0$ be the best policy learned by the agent during training under attack, the goal of the attack is to make the performance of $\pi_0$ in the environment $\mathcal{V}_{\mathcal{M}}^{\pi_0}$ as small as possible.

In this work, we want to find a black-box attack with limited computation resource that can achieve its goal with a limited budget. In the next section, we will formally formulate the attack problem.

## 4 Formulating Reward Poisoning Attack

Ideally, an optimal attacker would like to minimize the budget and the performance of $\pi_0$ at the same time, but such an attack may not exist as an attack may achieve less performance of $\pi_0$ with more budget. Therefore, it is more reasonable to define the optimal attacker with respect to a fixed budget. Formally, with the same amount of budget of $B$, $C$, the optimal attacker is the attack that can make the agent learn policies with the least performance. Let $T$ be the total training steps, we have the following optimization problem for the optimal attack.

$$\min_{\Delta^{t=1,\ldots,T}} V \text{ s.t. } \mathbb{E}[\mathcal{V}_{\mathcal{M}}^{\pi_0}] = V; \mathbb{E}[\sum_{t=1}^{T} \mathbb{1}[\Delta^t \neq 0]] = C; \max_t |\Delta^t| = B, \forall t \in [T]. \tag{1}$$

To solve the above optimization problem, one needs first to approximate the policy $\pi_0$ learned by the agent under the attack, then find the optimal attack strategy that makes the agent learn the worst policy. Both steps are hard in our black-box setting. To approximate $\pi_0$, the attacker needs knowledge of the environment

and agent. In our resource-constrained black-box attack setting, such information is not available at the beginning of training, and the attacker does not have sufficient memory or computing resources to learn that information based on the interaction between the agent and the environment at each step. For the second step, even in the white-box setting where the attacker has knowledge of the environment and the agent, the problem has already been shown to be hard to solve in the tabular MDP setting, not to mention when the state and action spaces are continuous in our DRL setting. Due to the above difficulties, instead of solving for the optimal or near-optimal attacks, we look for an efficient attacker that can satisfy the following conditions with small values of $(V, B, C)$ defined as below which can still be a major threat to the agent:

**Definition 4.1.** ($(V, B, C)$-efficient attack) We say an attack is $(V, B, C)$-efficient if it satisfies Equation 2 with $V$, $B$, and $C$:

$$\mathbb{E}[\mathcal{V}_{\mathcal{M}}^{\pi_0}] = V; \mathbb{E}[\sum_{t=1}^{T} \mathbb{1}[\Delta^t \neq 0]] = C; \max_t |\Delta^t| = B, \forall t \in [T]. \tag{2}$$

The expectation is with respect to the learning process in the environment under the attack strategy.

Note that similar formulations have been considered for data poisoning attacks in traditional RL settings (Jun et al., 2018; Liu & Shroff, 2019). To explain the technical details of our methods, we first introduce a measure for the distance $d : \mathcal{A} \times \mathcal{A} \to [0, 1]$ between two actions in Definition 4.2.

**Definition 4.2.** (distance between actions) For any two actions $a_1 \in \mathcal{A}, a_2 \in \mathcal{A}$, if the action space is discrete, then $d(a_1, a_2) = \mathbb{1}[a_1 \neq a_2]$. If the action space is continuous, then $d(a_1, a_2) = ||a_1 - a_2||_2/L$ where $L = \max_{a_1 \in \mathcal{A}, a_2 \in \mathcal{A}} ||a_1 - a_2||_2$ is the maximum $L_2$-norm difference between any two actions. Note that in both cases $d \in [0, 1]$.

Since we work in a black-box setting where no specific knowledge about the agent's learning algorithm is available, we adopt a general assumption for efficient learning during training:

**Assumption 4.3.** (Efficient learning algorithms) Let the total training steps be $T$. For any efficient learning algorithm, there exists $\delta \ll 1$, $\epsilon \ll 1$, and $p \ll 1$ such that given a stationary environment $\mathcal{M}$, the algorithm guarantees that after training for $T$ steps in $\mathcal{M}$, the following holds with probability at least $1 - p$: $\mathcal{V}_{\mathcal{M}}^{\pi_0} \geq \mathcal{V}_{\mathcal{M}}^{\pi^*} - \epsilon$ and $\sum_{t=1}^{T} d(a^t, \pi^*(s^t))/T \leq \delta$.

Intuitively, the assumption says that with a high probability, the agent should learn a near-optimal policy in the environment and take actions close to the optimal actions most of the time. We argue that such an assumption is reasonable. The value of $\epsilon$ represents how close the performance of the learned policy can be to that of the optimal policy with a high probability $1 - p$. Since the goal of a learning algorithm is to find a near-optimal policy in the environment with a high chance, the value of $\epsilon$ ought to be small for a small value of $p$. The value of $\delta$ represents how close the actions taken by a learning algorithm are to the optimal actions. To achieve the aforementioned learning goal efficiently, a learning algorithm needs to balance the exploration-exploitation trade-off and do a strategic exploration that explores sub-optimal actions less often, so we expect the value of $\delta$ is expected to be small. In practice, the choices for exploration strategies in current DRL algorithms include $E$-greedy and adding random noise on the action to explore, and both satisfy the assumption above. Note that in traditional RL settings, efficient learning algorithms are usually guaranteed to have such learning efficiency (Auer et al., 2002; Dong et al., 2019). For the DRL algorithms, we need to assume such guarantees on learning efficiency for predicting the behavior of the agent during training.

By Assumption 4.3, the learning algorithm can learn an $\epsilon$-optimal policy with a high probability when there is no attack, so we have the following definition for a trivial attack:

**Definition 4.4.** (Trivial attack) We say an attack is trivial if it is $(V, B, C)$-efficient with $V \geq (1-p) \cdot (V^* - \epsilon) + p \cdot V_{\min}$, where $V_{\min}$ is the minimal performance of a policy in $\mathcal{M}$. The reason is that when there is no attack, an agent can already learn a policy with performance greater than $V^* - \epsilon$ with probability at least $1 - p$ and otherwise $V_{\min}$ according to Assumption 4.3.

Based on Assumption 4.3, we propose the following attack framework that leverages this assumption for obtaining efficient attacks.

**Definition 4.5.** (Adversarial MDP Attack) The attacker constructs an adversarial MDP $\widehat{\mathcal{M}} = \{\mathcal{S}, \mathcal{A}, \mathcal{P}, \widehat{\mathcal{R}}, \mu\}$ which has a different reward function $\widehat{\mathcal{R}}$ compared to the true MDP. At time step $t$, let $r^t$ be the true reward generated by the environment. The attack samples $\hat{r}^t \sim \widehat{\mathcal{R}}(s^t, a^t)$ and injects corruption as $\Delta^t = \hat{r}^t - r^t$. In the end, the reward observed by the agent at step $t$ is $\hat{r}^t$.

Under the adversarial MDP attack, the agent trains in the adversary environment $\widehat{\mathcal{M}}$ constructed by the attacker. Since the agent still trains in a stationary adversarial environment, Assumption 4.3 which is about the agent's learning behavior in a stationary environment is valid. So the general behavior of the agent is predictable under the adversarial MDP attack. Next, we will propose instances that solve equation 2 efficiently.

# 5 Poisoning Attack Methods

In this section, we design methods to construct adversarial MDP attacks under the $(T, \epsilon, \delta)$-efficient learning algorithm assumption and analyze their efficiency according to Equation 2. We assume the number of training steps as the value of $T$, therefore the inequalities in Assumption 4.3 hold with $\epsilon \ll 1$ and $\delta \ll 1$ after $T$ training steps in any stationary environment. First, we define a naive baseline attack and show that it is inefficient, which motivates us to propose the principles to make an adversarial MDP attack efficient.

**Uniformly random time (UR) attack.** The attacker randomly corrupts the reward at a timestep $t$ with a fixed probability $p$ and amount $\Delta$ regardless of the current state and action. Formally, the attack strategy of the UR attacker at time $t$ is $\Delta^t = \Delta$ with probability $p$, otherwise $\Delta^t = 0$. The UR attack requires very limited computation resources as it only needs to sample from a Bernoulli distribution with a fixed probability $p$ at every training step. For the efficiency of the attack, it satisfies the attack problem in Equation 2 with $(V, B, C)$ as below:

**Theorem 5.1.** *For the UR attack parameterized with attack probability $\kappa$, per-step corruption $\Delta$, let $L^\pi := \sum_s \mu^\pi(s)$ be the expected length of an episode before termination for an agent following the policy $\pi$. The optimal policy under the adversarial environment is $\hat{V}^* = \arg\max_\pi \mathcal{V}^\pi_{\mathcal{M}} - \Delta \cdot \kappa \cdot L^\pi$. The UR attack satisfies equation 2 with $B = |\Delta|$, $C = \kappa \cdot T$, and $V \leq (1 - p) \cdot \max_{\pi: \mathcal{V}^\pi_{\mathcal{M}} - \Delta \cdot \kappa \cdot L^\pi \geq \hat{V}^* - \epsilon} \mathcal{V}^\pi_{\mathcal{M}} + p \cdot V^*$.*

The proof for all theorems and lemmas in this section can be found in the appendix. Note that $L^\pi$ is independent of the reward function, so it cannot be influenced by the attack. In the following lemma, we identify types of environments and $\Delta$ where the UR attack is trivial. Note that these cases are common and have been considered in related work for poisoning attacks against DRL (Sun et al., 2020).

**Lemma 5.2.** *The UR attack is trivial if: (i) $L^\pi$ is the same for all policies, or (ii) $\mathcal{V}^\pi_{\mathcal{M}}$ is monotonically increasing (or decreasing) with $L^\pi$, and $\Delta$ is negative (or positive).*

Lemma 5.2 shows that the UR attack can be trivial in some common cases, and we experimentally find that the efficiency of the UR attack is limited for attacking different learning algorithms in diverse environments. We believe the main reason making the UR attack inefficient is that the $\epsilon$-optimal policies in the adversarial environment constructed by the UR attack have high performance in the true environment. As a result, the agent will learn a policy of high performance under the attack. In this case, let $\Pi_\epsilon = \{\pi | \mathcal{V}^{\hat{\pi}^*}_{\widehat{\mathcal{M}}} - \mathcal{V}^\pi_{\widehat{\mathcal{M}}} \leq \epsilon\}$ be the set of $\epsilon$-optimal policies in the adversarial environment. Then for any $\pi_1 \in \Pi_\epsilon$ and some $\pi_2 \in \Pi \setminus \Pi_\epsilon$, we have $\mathcal{V}^{\pi_1}_{\widehat{\mathcal{M}}} > \mathcal{V}^{\pi_2}_{\widehat{\mathcal{M}}}$ and $\mathcal{V}^{\pi_1}_{\mathcal{M}} \ll \mathcal{V}^{\pi_2}_{\mathcal{M}}$. In other words, for some policies $\pi_1, \pi_2$ as described above, their relative performance is very different in the true and adversarial environment. In the cases given by Lemma 5.2, for any policy $\pi_1 \in \Pi_\epsilon$ and $\pi_2 \in \Pi \setminus \Pi_\epsilon$, we have either $\mathcal{V}^{\pi_1}_{\widehat{\mathcal{M}}} - \mathcal{V}^{\pi_2}_{\widehat{\mathcal{M}}} = \mathcal{V}^{\pi_1}_{\mathcal{M}} - \mathcal{V}^{\pi_2}_{\mathcal{M}}$ or $\mathcal{V}^{\pi_1}_{\mathcal{M}} > \mathcal{V}^{\pi_2}_{\mathcal{M}}$, resulting in attack inefficiency. Beyond these cases, the UR attack is still inefficient, and we believe the reason is that the attack uniformly applies perturbation to all state-action pairs. To achieve the aforementioned conditions on $\pi_1$ and $\pi_2$, the difference in reward given by the true and adversarial environment $\mathbb{E}[\widehat{\mathcal{R}}(s, a) - \mathcal{R}(s, a)]$ should be high for some state-action pairs at least. Under the UR attack, the difference in reward for every state-action pair is always $\mathbb{E}[\widehat{\mathcal{R}}(s, a) - \mathcal{R}(s, a)] = \Delta \cdot \kappa = B \cdot C/T$. Note that the difference is bound by $\mathbb{E}[\widehat{\mathcal{R}}(s, a) - \mathcal{R}(s, a)] \leq B$ according to the definition of $B$. To achieve the highest possible value of $\mathbb{E}[\widehat{\mathcal{R}}(s, a) - \mathcal{R}(s, a)]$, the UR attack requires $C = T$ which is too large. An efficient attacker should achieve this with a much smaller requirement

of $C$. This can happen if the attack only perturbs the actions far from the optimal actions of the adversarial environment at every state. Since the agent will explore optimal actions for most training steps according to Assumption 4.3, the attacker does not need to apply perturbation for those training steps. As a result, the attacker only requires a low value for $C$ while it perturbs the sub-optimal actions in the adversarial environment as much as possible, i.e., by the value of $B$. In summary, we have two principles for designing efficient adversarial MDP attacks.

**Efficient adversarial MDP attack principles:**

1. The $\epsilon$-optimal policies in the adversarial environment have low performance in the true environment, and

2. The reward functions for the adversarial and true environments have the same output at every state for the actions whose distance to the actions given by the optimal policy in the adversarial environment is small.

Next, we propose ways to construct an adversarial MDP attack following the efficient attack principles and analyze their efficiency.

**Action evasion (AE) attack.** The high-level idea behind the AE attack is to make the agent believe that at each state some actions are not optimal. Then it will learn a policy that never takes these actions in the corresponding states. In the RL setting, an agent needs to take optimal actions at every state to maximize its total reward. Failing to take optimal actions at any state will lead to a sub-optimal outcome. If the attacker arbitrarily selects a sub-set of actions to attack, then the sub-set of actions will contain the optimal action at some states with a high probability. Therefore the agent will not learn optimal actions at these states leading to sub-optimal results. In many cases, such sub-optimality can be high. For example, in a challenging maze problem, the agent needs to find a way from the entrance to the exit. There can be some crossroads where there is only one correct direction that leads to the exit. At every location in the maze, the attacker makes the agent not take a random action. Then with a high probability, in at least one crossroad the random action can be the correct action. As a result, the agent will learn a policy that does not select the correct direction at that crossroad so it can never reach the exit, which corresponds to the worst outcome. With this intuition, the policy learned by the agent that will never take some actions at every state can have low performance. Following this idea, the AE attack applies penalties on reward whenever the agent takes certain actions at every state. Formally, the AE attack is defined as follows.

**Definition 5.3.** (Action evasion attack) The attack specifies an arbitrary policy $\pi^\dagger$, a constant $\Delta > 0$, and a radius $r \in (0,1)$. At time $t$, the attack strategy is $\Delta^t = -\Delta \cdot \mathbb{1}\{d(a^t, \pi^\dagger(s^t)) < r\}$.

Note that for the discrete action spaces, any value of $r \in (0,1)$ will lead to the same attack that only applies perturbation when the action taken by the agent is the same as the one given by $\pi^\dagger$. The computation resource required by the AE attack is the memory to store the policy $\pi^\dagger$ and the computation power to calculate the output action of the policy for an input state at every training step. For ease of analysis, we define the distance between any two policies.

**Definition 5.4.** ($r$-Distance between two policies) For a value $r \in (0,1)$, the $r$-distance between two policies is defined as

$$D_r(\pi_1, \pi_2) = \mathbb{E}_{s \sim \mu_1^\pi} \mathbb{1}\{d(\pi_1(s), \pi_2(s)) \geq r\}.$$

By definition, the value of the distance between two policies is bound by $D_r(\pi_1, \pi_2) \in [0, L^{\pi_1}]$. We say $\pi_2$ is in the $r$-neighborhood of $\pi_1$ if $D_r(\pi_1, \pi_2) = 0$, and $\pi_2$ is $r$-far from $\pi_1$ if $D_r(\pi_1, \pi_2) = L^{\pi_1}$. Also note that if $D_r(\pi_1, \pi_2) = 0$, then $D_r(\pi_2, \pi_1) = 0$. Though in general, $D_r$ is not symmetric.

The efficiency of the AE attack satisfies the following:

**Theorem 5.5.** *Let* $\hat{\pi}^* = \arg\max_{\pi : D_r(\pi, \pi^\dagger) = L^\pi} \mathcal{V}_{\mathcal{M}}^\pi$. *If* $\Delta$ *satisfies* $\mathcal{V}^{\hat{\pi}^*} - \epsilon > \max_{\pi : D_r(\pi, \pi^\dagger) < L^\pi} \mathcal{V}_{\mathcal{M}}^\pi - \Delta \cdot (L^\pi - D_r(\pi, \pi^\dagger))$, *then the AE attack satisfies Equation 2 with* $V \leq (1-p) \cdot \mathcal{V}_{\mathcal{M}}^{\hat{\pi}^*} + p \cdot V^*$, $B = |\Delta|$, $C \leq (1-p) \cdot \frac{\delta}{\min_s \{d(\hat{\pi}^*(s), \pi^\dagger(s)) - r\}} \cdot T + p \cdot T$. *In the discrete action space case, the bound on* $C$ *can be rewritten as* $C \leq (1-p) \cdot \delta \cdot T + p \cdot T$.

For the AE attack, if the per-step perturbation $\Delta$ is sufficiently large (i.e., satisfies the constraint above), the $\epsilon$-optimal policies in the adversarial environment are always $r$-far from $\pi^\dagger$. Since the actions that will

be perturbed by the attacker are all sub-optimal actions that the agent will not frequently explore, the number of training steps to be corrupted by the attack is limited. Theorem 5.5 implies that the performance degradation (the gap between $V$ and $V^*$) induced by the AE attack depends on the highest performance of the policies that are $r$-far from $\pi^\dagger$. According to our discussion before, in many cases, the policies that never take some actions in every state have low performance, then the performance degradation will be large. Though we note that there are cases where such policies have sub-optimal yet decent performance.

Consider environments where policies of high performance have very distinct behaviors. In these environments, the sub-optimality in the outcome induced by taking a non-optimal action at any state is limited, as one can still find a sub-optimal yet high-performing policy after taking a non-optimal action. As a result, policies failing to take optimal actions in some states do not necessarily have low performance. For example, consider a simple maze problem where there is no wall. The AE attack can make the agent believe that at each position, it should not take a particular direction. Since at each position, there are always paths that reach the exit without moving in that particular direction, the agent may still find a path to the exit in a maze. Such a path may take more steps to reach the exit compared to the optimal path, but its sub-optimality in performance is limited compared to the paths which cannot reach the exit. To handle these cases, we propose another adversarial MDP attack.

**Action inducing (AI) attack.** The high-level idea of the AI attack is to make the agent believe that some random actions at every state are optimal. Then the agent will learn a policy that always takes random actions at every state, and such random behavior usually leads to poor performance in practice. In the example of the simple maze problem where there is no wall, a policy that randomly walks around in the maze is very unlikely to reach the exit, leading to the worst performance. To achieve this, the attacker penalizes the agent whenever it takes an action different from the ones suggested by the random actions. Formally, the AI attack is defined as follows.

**Definition 5.6.** (Action inducing attack) The attack randomly specifies policy $\pi^\dagger$, a constant $\Delta < 0$, and a radius $r$. At time $t$, the attack strategy is $\Delta^t = -\Delta \cdot \mathbb{1}\{d(a^t, \pi^\dagger(s^t)) \geq r\}$.

Note that for the discrete action spaces, any value of $r \in (0, 1)$ will lead to the same attack that only applies perturbation when the action taken by the agent is different from the one given by $\pi^\dagger$. Same as the case of the AE attack, the computation resources required by the AI attack is the memory to store the policy $\pi^\dagger$ and the computation power to calculate the output action of the policy for an input state at every training step. The efficiency of the AI attack satisfies:

**Theorem 5.7.** *If $\Delta$ satisfies $\max_{\pi:D_r(\pi,\pi^\dagger)=0} \mathcal{V}_{\mathcal{M}}^\pi - \epsilon > \max_{\pi:D_r(\pi,\pi^\dagger)>0} \mathcal{V}_{\mathcal{M}}^\pi - \Delta \cdot D_r(\pi,\pi^\dagger)$, then adversarial optimal policy under the attack is $\hat{\pi}^* = \arg\max_{\pi:D_r(\pi,\pi^\dagger)=0} \mathcal{V}_{\mathcal{M}}^\pi$. In this case, the AI attack satisfies Equation 2 with $V \leq (1-p) \cdot \mathcal{V}_{\mathcal{M}}^{\hat{\pi}^*} + p \cdot V^*$, $B = |\Delta|$, and $C \leq (1-p) \cdot \frac{\delta}{\min_s \{r - d(\hat{\pi}^*(s), \pi^\dagger(s))\}} \cdot T + p \cdot T$. In the discrete action space case, the bound on $C$ can be rewritten as $C \leq (1-p) \cdot \delta \cdot T + p \cdot T$.*

Basically, with high enough per-step perturbation, the $\epsilon$-optimal policies in the adversarial environment are always in the $r$-neighborhood of $\pi^\dagger$. Since the optimal actions are among the actions that the AI attack will not perturb, the attacker only needs to perturb a limited number of steps when the agent explores sub-optimal actions. Theorem 5.7 implies that the performance degradation induced by the AI attack depends on the highest performance of the policies that are in the $r$-neighborhood of $\pi^\dagger$, which is a randomly generated policy. Such policies usually have low performance as we discussed before, so the performance degradation is likely to be large. In Section 6, we experimentally test the performance of different attacks.

## 6 Experiments

We evaluate our attack methods from Section 5 for poisoning the training process with state-of-the-art DRL algorithms in both the discrete and continuous settings. We consider learning in environments typically used for assessing the performance of the DRL algorithms in the literature.

**Learning algorithms and environments:** We consider 4 common Gym environments (Brockman et al., 2016) in the discrete case: CartPole, LunarLander, MountainCar, and Acrobot, and 4 continuous cases: HalfCheetah, Hopper, Walker2d, and Swimmer. The DRL algorithms in the discrete setting are: dueling

deep Q learning (Duel) (Wang et al., 2016) and double dueling deep Q learning (Double) (Van Hasselt et al., 2016) while for the continuous case we choose: deep deterministic policy gradient (DDPG), twin delayed DDPG (TD3), soft actor critic (SAC), and proximal policy optimization (PPO). The implementation of the algorithms is based on the spinningup project (Achiam, 2018). The number of training steps $T$ is set to be large enough so that the learning algorithm can converge within $T$ time steps without the attack.

**Efficiency of the attacks:** First, we want to show that our black-box attacks are efficient in diverse representative learning scenarios, i.e., they make the agent learn policies of low performance under limited budgets. As mentioned in Section 4, we consider an additional constraint on the total perturbation applied during an episode $E$. Despite this additional restriction on attack power, we show that our attacks remain efficient under a limited budget of $E$. We study the influence of different values of $E$ on the efficiency of the attack in Appendix A.

To make the comparison fair, we let all the attacks share the same budget of $B$ and put hard limits on budgets of $C$, and $E$. Then the attack that achieves a smaller value of $V$, the performance of the best policy learned by the agent, is more efficient. The hard limit on budgets means that the attacker cannot apply perturbation if doing so according to its attack strategy will break the limit. Formally, if at time $t$ in episode $e$, $1 + \sum_{\tau=0}^{t-1} \mathbb{1}[|\Delta^\tau > 0] > C$ or $|\Delta| + \sum_{\tau \in t_e} |\Delta^\tau| > E$, then the attacker applies no corruption at that time step. The hard limit on the budget represents a more realistic attack scenario as the attack can be easily detected if it uses budgets more than the hard limits. We next described our choices on the values of $B$, $C$, and $E$, attack parameters $r$ and $\Delta$, and how we measure $V$ in our experiments.

**Determing $B$.** Let $V_{\max}$, $V_{\min}$ be the highest and lowest expected reward a policy can get in an episode. We note that $V_{\max} - V_{\min}$ represents the maximum environment-specific net reward an agent can get during an episode, and $\frac{(V_{\max} - V_{\min})}{L_{\max}}$ represents the range of average reward at a time step for an agent. We set $B$ to be higher than $\frac{(V_{\max} - V_{\min})}{L_{\max}}$ as a low value of $B$ may not influence the learning process for any attack. A similar choice has also been considered in simpler tabular MDP settings (Zhang et al., 2020b).

**Determing $E$.** We restrict the value of $E$ to be less than $V_{\max} - V_{\min}$ to ensure that the perturbation in each episode is less than the net reward.

**Determing $C$.** We want the attack to corrupt as few training steps as possible, so we set the value of $C/T \ll 1$. Since this is usually the most important budget, we test the efficiency of the attack under different values of $C/T \in [0.005, 0.2]$.

**Determing Attack parameters.** For the choice of $r$ in the continuous action spaces, we choose $r$ from a moderate range $r \in [0.3, 0.75]$ for each learning scenario. In Appendix A, we study the influence of $r$ on the attack and show that the attack can work well within a wide range of values for $r$. For all attacks, we have $|\Delta| = B$ by construction. The sign of $\Delta$ of AE and AI attack are given by Definition 5.3 and 5.6. For the UR attack, we test both cases with $\Delta = B, -B$ and show the best result. For the choice of $\pi^\dagger$, according to Definition 5.6, we randomly generate $\pi^\dagger$ for the AI attack. For the AE attack, since the attack works in the black-box setting, $\pi^\dagger$ is also randomly generated. We study the influence of the choice of $\pi^\dagger$ for the AE attack in Appendix A.

**Measuring $V$.** To evaluate the value of $V$, we test the empirical performance of the learned policy after each epoch and report the highest performance as the value of $V$. We repeat each experiment 10 times and report the average value. We report the variance of results in the appendix.

**Results.** Fig 1 compares the efficiency of the UR, AE, and AI attacks. We also show the performance of the agent without attack. We find that in most cases, the UR attack is the least efficient as it yields learned policies with similar performance as without attack. For the AE and AI attacks, we find that the attacks are usually efficient in the learning scenarios we test despite a few failing cases like learning in LunarLander environments with the dueling DQN algorithm. The reason can be that our assumptions for learning algorithms don't hold in this case. We further notice that in different learning scenarios, the relative efficiency of the two attacks is different. For example, in the case of learning in the HalfCheetah environment by DDPG algorithm, the AE attack is much more efficient than the AI attack; for learning

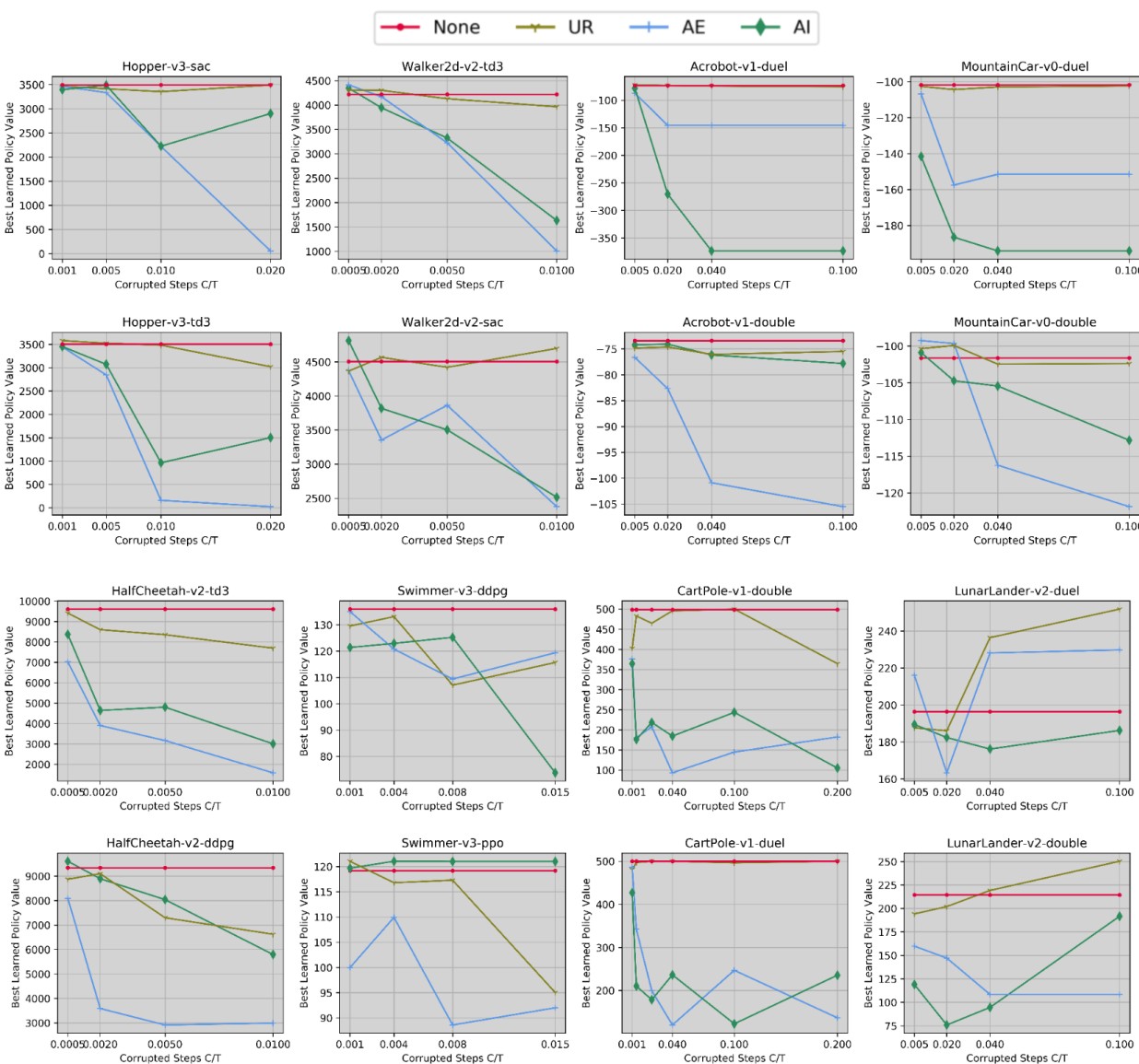

Figure 1: The highest performance of the policy learned by different learning algorithms in different environments under our attack methods.

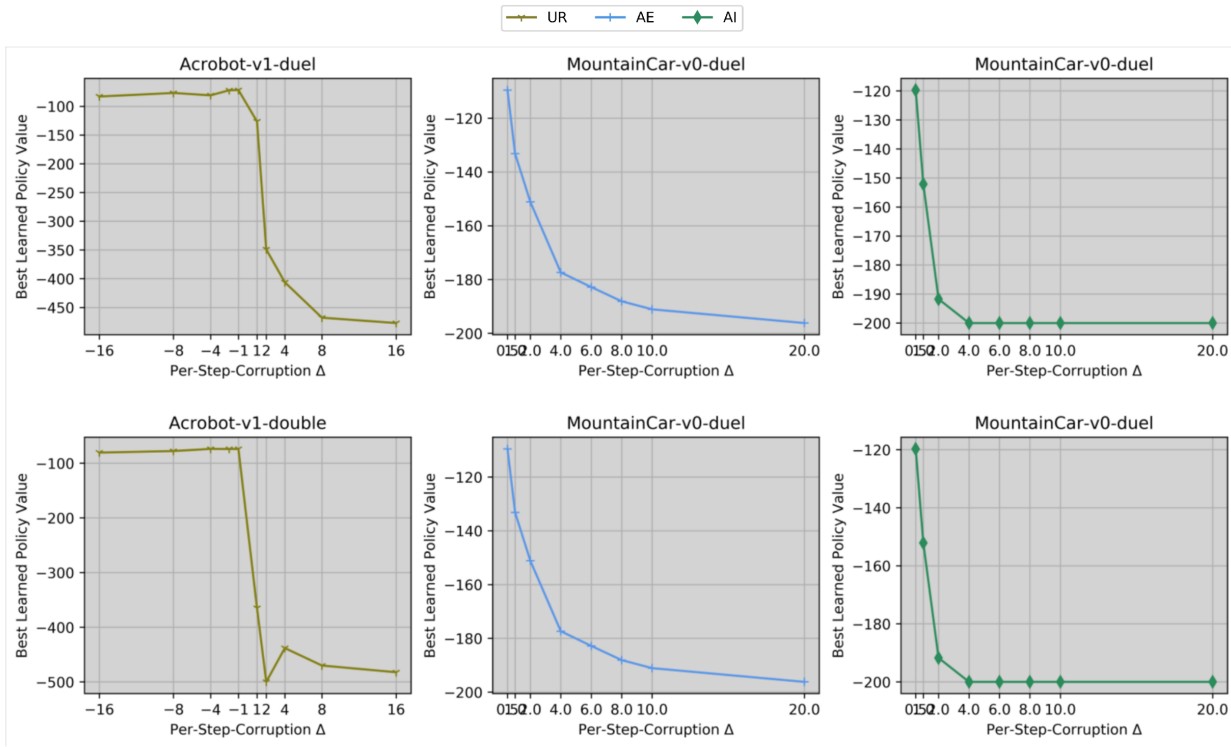

Figure 2: Experimental observations validating implications 1,2,4, and 6 from section 5.

in the MountainCar environment with the dueling DQN algorithm, the AI attack is more efficient. These observations vindicate our choice of proposing separate attacks to handle different cases.

**Influence of $\Delta$ on different attacks:** Next, we study the effect of the per-step corruption $\Delta$ on the attack efficiency. To exclude the effects of attack constraints, and the choice of $r$ on the policy performance $V$, we do not consider a hard limit on $C$ or $E$ and only study discrete action spaces. Fig 2 shows our results. For the UR attack, we find that when $\Delta$ is negative, the UR attack cannot influence the learning efficiency of the agent, which is expected by Lemma 5.2. For the AE and AI attacks, we find that when $\Delta$ is small, the influence of the attack on $V$ is limited. As the value of $\Delta$ increases, the influence on $V$ becomes more significant. When $\Delta$ is large enough, the value of $V$ does not change much as $\Delta$ increases. As pointed out by Theorem 5.5, 5.7, such a phenomenon indicates that when $\Delta$ is big enough, the agent will find the near-optimal policies among the ones that are far from (AE) or in the neighborhood (AI) of $\pi^{\dagger}$.

**Verification of the efficient learning assumption:** Here we verify one key insight behind the efficiency of the AE and AI attack: with sufficient values of $\Delta$, the attack will not apply perturbation when the agent takes actions close to the optimal actions in the adversarial environment, and this happens for most of the training steps according to Assumption 4.3. This insight explains why the attack can work without requiring a large budget for $C$. Fig 3 shows the result where we run the AE and AI attacks on the MountainCar environment with different learning algorithms. We observe in Fig 3 that after a few initial epochs, the agent rarely takes the actions that the attack will perturb in the states, suggesting that our insight is correct in this empirical case.

# 7 Conclusion and Limitations

In this work, we study the security vulnerability of DRL algorithms against training time attacks. We designed a general, parametric framework for reward poisoning attacks and instantiated it to create several efficient attacks. We provide theoretical analysis for the efficiency of our attacks. Our detailed empirical

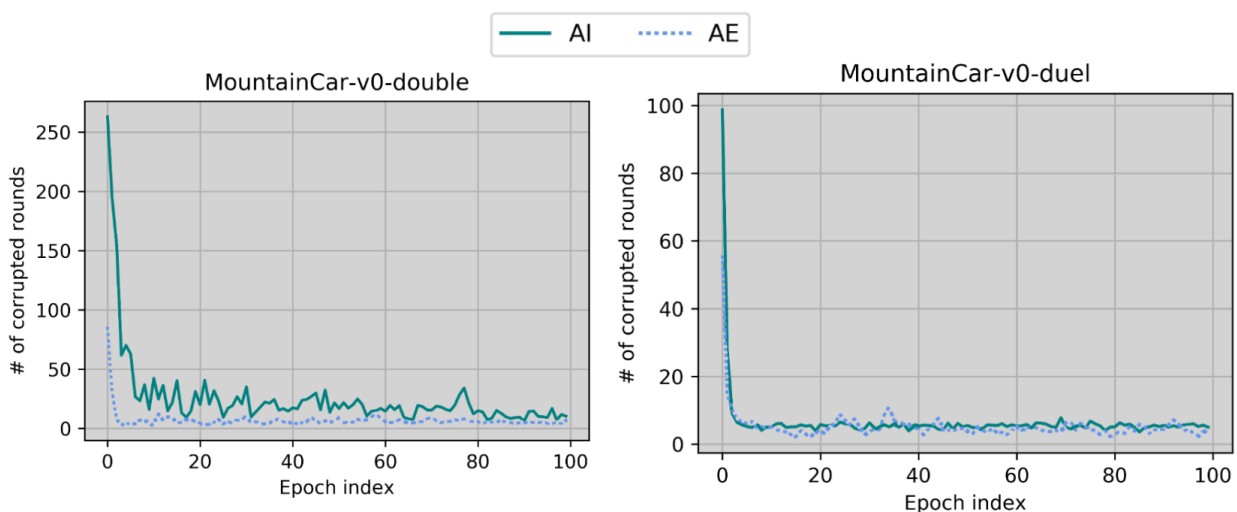

Figure 3: Experimental observations validating implications 3 and 5 from section 5

evaluation confirms the efficiency of our attack methods pointing to the practical vulnerability of popular DRL algorithms. We introduce this attack in the hopes that highlights that practical attacks exist against DRL. Our goal is to spawn future work into developing more robust DRL learning methods.

Our attacks have the following limitations: (i) not applicable for other attack goals, e.g, to induce a target policy, (ii) cannot find the optimal attacks, and (iii) do not cover state poisoning attacks.

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

## A   Experiments details and additional experiments

**Influence of $\pi^\dagger$ on the AE attack**: Recall that the AE attack can make the agent believe that at each state some actions are not optimal. If such actions are associated with high future rewards, the policy learned by the agent may have even worse performance as it will never take such actions. Following this intuition, the AE attack may have higher efficiency if attacks with a policy $\pi^\dagger$ of higher performance compared to a random policy. Note that this requires the attacker to have access to policies of high performance of the environment before agent training begins, which is not available to the black-box attack setting considered in the main paper. Therefore, to study the influence of $\pi^\dagger$ on the AE attack, we remove the constraint on attacker's knowledge of the environment that here the attacker has access to some high performing policies in the environment. Formally, we consider two types of $\pi^\dagger$: (1) expert policy whose performance matches the best policy an efficient learning algorithm can learn (2) medium policy whose performance is in the middle between an expert policy and a random policy. With the same setup, we compare the AE attack in the three cases with a random, medium, and expert policy respectively. The results in Fig 4 show that the AE attack with $\pi^\dagger$ of higher performance usually has much more influence on the attack with a random policy in most learning scenarios.

**Influence of $r$ on attack's efficiency**: Here we study the effect of the hyperparameter $r$ on the efficiency of our attacks. Recall that for AE or AI attack with a policy $\pi^\dagger$, a reward penalty of $\Delta$ will be applied if the agent selects an action at state $s$ with distance to $\pi^\dagger(s)$ less than or greater than $r$ respectively. From Theorem 5.5, under the efficient learning algorithm assumption, we know that with a sufficient value of $\Delta$, the efficiency of the attack on $V$ satisfies $V \leq \max_{\pi:D_r(\pi,\pi^\dagger)=L^\pi} \mathcal{V}_\mathcal{M}^\pi$. So if the value of $r$ decreases, the set of policies satisfy $\pi:D_r(\pi,\pi^\dagger)=L^\pi$ will increase, then the value of $V$ will also increase, making the attack less efficient. Similarly, from theorem 5.7, the value of $V$ for the AI attack satisfies $V \leq \max_{\pi:D_r(\pi,\pi^\dagger)=0} \mathcal{V}_\mathcal{M}^\pi$.

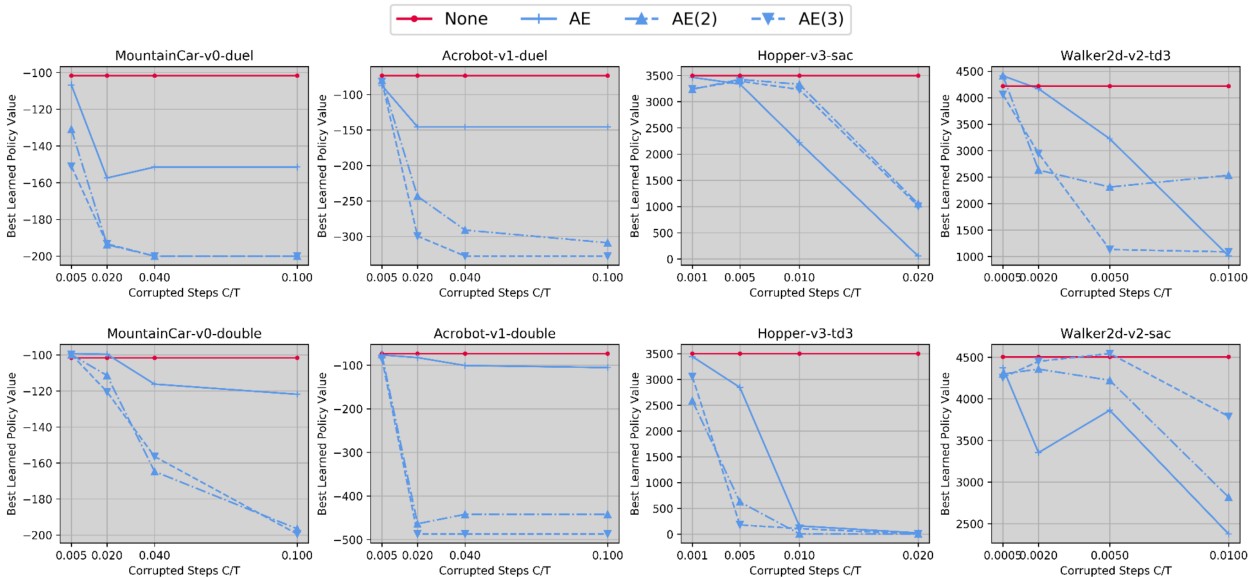

Figure 4: Comparison between the AE attacks with $\pi^\dagger$ of different performances. $\pi^\dagger$ is a random policy in 'AE', medium policy in 'AE(2)', and expert policy in 'AE(3)'

Then if $r$ is increased, the set of policies satisfy $\pi : D_r(\pi, \pi^\dagger) = 0$ will increase, so the value of $V$ will increase. Note that the above statements are based on the assumption of the efficient learning algorithm. If the value of $r$ is extremely small (close to 0) or large (close to 1), this assumption may break so we only test the value of $r$ in a relatively moderate range. We test in an example case that learns in the Hopper environment with TD3 algorithm. The value of $\Delta$ is set to be 25, and there is no hard limit on $C$. The results in Fig 5 verify the above statement in the example learning scenarios.

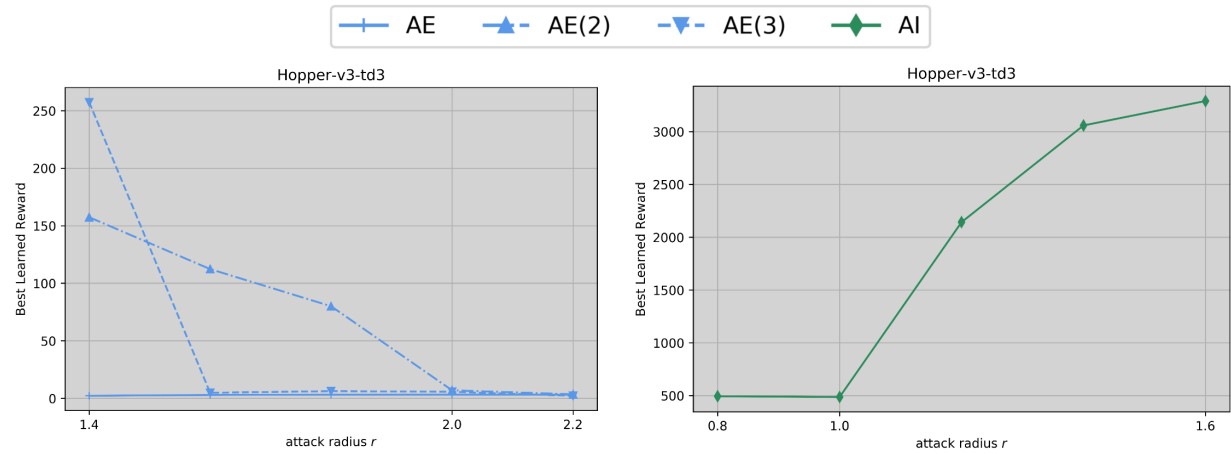

Figure 5: Influence of $r$ on different attack methods

**Influence of $E$ on attack's efficiency** Recall in the experiments, to make the setting more realistic, we consider an additional constraint on the attack where its total amount of perturbation for each episode is no greater than $E$. Here we study what is the influence of $E$ on the efficiency of the attack experimentally. We test in two example environments where the agent learns in Acrobot and Mountaincar environments with dueling DQN algorithms. The parameters and hard limits on the attack are set to be the same as in Table 1 except for the value of $E$. The results in Fig 6 show that unless the value of $E$ is too small, e.g, when

$E$ is greater than the per-episode net-reward range as considered in Section 6, the efficiency on $V$ does not change much for different values of $E$.

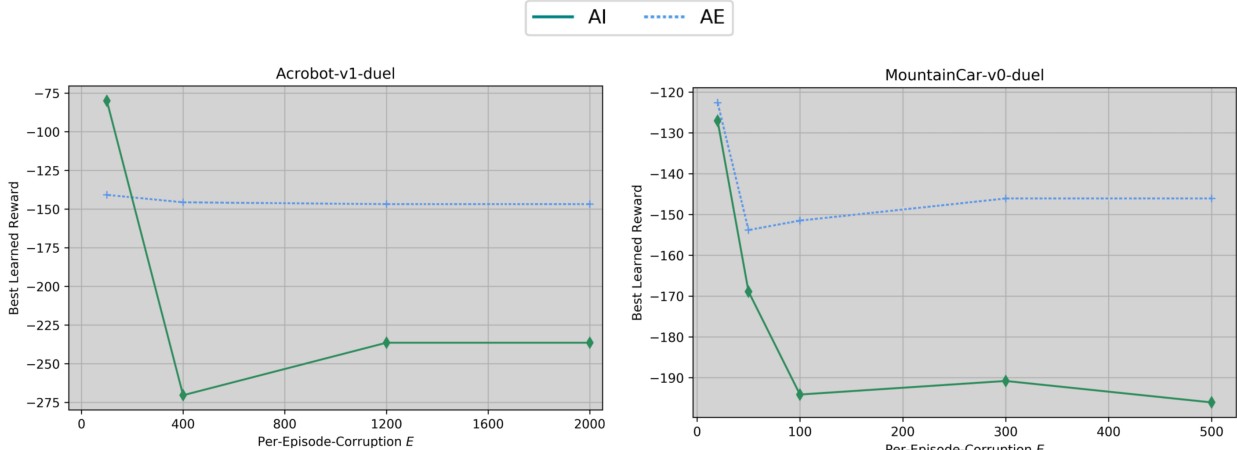

Figure 6: Influence of $E$ on the efficiency of the attack

**Detailed parameters used in the table**: The hyper-parameters for the learning algorithms can be found in the codes. The parameters for the setup of the experiments are given in Table 1. In Table 2 We provide the value of $V_{\min}$, $V_{\max}$, and $L_{\max}$ for each environment we use to determine the constraints on the attack. These values are given by either the setup of the environment or empirical estimation. For example, in MountainCar-v0, $L_{\max} = 200$ and $V_{\min} = -200$ are given by the setup of the environment directly, as an episode will be terminated after 200 steps, and the reward is $-1$ for each step. $V_{\max} = -100$ is empirically estimated by the highest reward given by the best policy learned by the most efficient learning algorithm.

Table 1: Parameters for experiments

| Environment | $T$ | $B$ | $E$ | $r(\text{AE})$ | $r(\text{AI})$ |
|---|---|---|---|---|---|
| CartPole | 80000 | 5 | 500 | / | / |
| LunarLander | 120000 | 4 | 800 | / | / |
| MountainCar | 80000 | 2.5 | 200 | / | / |
| Acrobot | 80000 | 4 | 500 | / | / |
| HalfCheetah | 600000 | 42 | 6300 | 2 | 1.5 |
| Hopper | 600000 | 25 | 2500 | 2 | 1.1 |
| Walker2d | 600000 | 25 | 2500 | 2.2 | 1.5 |
| Swimmer | 600000 | 0.8 | 80 | 2.2 | 1.0 |

Table 2: Values used to determine the constraints on attack

| Environment | $L_{\max}$ | $V_{\max}$ | $V_{\min}$ | $V_{\max} - V_{\min}$ | $\frac{V_{\max} - V_{\min}}{L_{\max}}$ |
|---|---|---|---|---|---|
| CartPole | 500 | 500 | 0 | 500 | 1 |
| LunarLander | 1000 | 200 | -1000 | 1200 | 1.2 |
| MountainCar | 200 | -100 | -200 | 100 | 0.5 |
| Acrobot | 500 | -100 | -500 | 400 | 0.8 |
| HalfCheetah | 1000 | 12000 | 0 | 12000 | 12 |
| Hopper | 1000 | 4000 | 0 | 4000 | 4 |
| Walker2d | 1000 | 5000 | 0 | 5000 | 5 |
| Swimmer | 1000 | 120 | 0 | 120 | 0.12 |

Table 3: The values in the table are the variance of the performance of the best learned policy over 10 runs in identical settings. The values in the brackets are the corresponding average value that is reported in figure 1.

| Environment-Learning algorithm-Attack | C = 0.001 | C = 0.005 | C = 0.01 | C = 0.02 |
|---|---|---|---|---|
| Hopper-sac-UR | 106(3468) | 114 (3414) | 296(3355) | 90(3492) |
| Hopper-sac-AE | 125 (3463) | 165 (3335) | 1134 (2220) | 113 (55) |
| Hopper-sac-AI | 107(3397) | 81(3489) | 1118(2224) | 628(2901) |
| Hopper-td3-UR | 147(3580) | 212(3524) | 215(3482) | 1032(3021) |
| Hopper-td3-AE | 133(3443) | 895(2845) | 286(159) | 49(23) |
| Hopper-td3-AI | 87(3452) | 860(3072) | 206(963) | 690(1501) |

# B Proof for theorems and lemmas

**Proof for Theorem 5.1** By the definition of the attack, we have $B = |\Delta|$, $C = p \cdot T$. The policy value of a policy $\pi$ under the adversarial environment constructed by the AE attack with $\Delta$, $r$ and $\pi^\dagger$ satisfies: $\mathcal{V}^\pi_{\widehat{\mathcal{M}}} = \mathcal{V}^\pi_{\mathcal{M}} - \Delta \cdot L^\pi$. Then in the adversarial environment, an $\epsilon$-optimal policy satisfies $\mathcal{V}^\pi_{\mathcal{M}} - \Delta \cdot L^\pi \geq \hat{V}^* - \epsilon$ where $\hat{V}^* = \max_\pi \mathcal{V}^\pi_{\mathcal{M}} - \Delta \cdot L^\pi$.

Let $G$ be the event that the agent learns an $\epsilon$-optimal policy in the adversarial environment. When $G$ is true, the performance on the true environment of the policy learned by the agent is bound by $\max_{\pi:\mathcal{V}^\pi_{\mathcal{M}} - \Delta \cdot L^\pi \geq \hat{V}^* - \epsilon} \mathcal{V}^\pi_{\mathcal{M}}$. When $G$ is false, the performance is bound by $V^*$. Combining both cases, we have $V \leq \max_{\pi:\mathcal{V}^\pi_{\mathcal{M}} - \Delta \cdot L^\pi \geq \hat{V}^* - \epsilon} \mathcal{V}^\pi_{\mathcal{M}}$.

**Proof for Theorem 5.2** In the first case, $L^\pi$ is the same for any policy $\pi$. Then for any two policies $\pi_1$ and $\pi_2$, the difference between their policy values in the true and adversarial environments are the same $\mathcal{V}^{\pi_1}_{\mathcal{M}} - \mathcal{V}^{\pi_2}_{\mathcal{M}} = \mathcal{V}^{\pi_1}_{\widehat{\mathcal{M}}} - \mathcal{V}^{\pi_2}_{\widehat{\mathcal{M}}}$. Therefore, $\epsilon$-optimal policies are the same in both environments, so under the efficient learning assumption, with probability at least $1 - p$, the agent will learn a policy whose policy value is at least $V^* - \epsilon$. Then the value of $V$ is at least $(1 - p) \cdot (V^* - \epsilon) + p \cdot V_{\min}$ where $V_{\min}$ is the minimal performance of a policy in $\mathcal{M}$. So the attack is trivial.

In the second case, the policy value $\mathcal{V}^\pi_{\mathcal{M}}$ is monotonic increasing or decreasing with $L^\pi$, and $\Delta$ is negative or positive. For any two policies $pi_1$ and $\pi_2$ such that $\pi_1$ has a higher policy value than $\pi_2$: $\mathcal{V}^{\pi_1}_{\mathcal{M}} - \mathcal{V}^{\pi_2}_{\mathcal{M}} > 0$, then in the adversarial environment, $\pi_1$ has a even higher policy value than $\pi_2$: $\mathcal{V}^{\pi_1}_{\widehat{\mathcal{M}}} - \mathcal{V}^{\pi_2}_{\widehat{\mathcal{M}}} > \mathcal{V}^{\pi_1}_{\mathcal{M}} - \mathcal{V}^{\pi_2}_{\mathcal{M}}$. So the $\epsilon$-optimal policies in the adversarial environment are a subset of $\epsilon$-optimal policies in the true environment. Then the attack is trivial for the same reason as the above case.

**Proof for Theorem 5.5**

*Proof.* The policy value of a policy $\pi$ under the adversarial environment constructed by the AE attack with $\Delta$, $r$ and $\pi^\dagger$ satisfies: $\mathcal{V}^\pi_{\widehat{\mathcal{M}}} = \mathcal{V}^\pi_{\mathcal{M}} - \Delta \cdot (L^\pi - D_r(\pi, \pi^\dagger))$. Note that for any policy that is far from $\pi^\dagger$, i.e. $D_r(\pi, \pi^\dagger) = L^\pi$, its policy value is the same in both true and adversarial environments. If the following is satisfied: $\mathcal{V}^{\hat{\pi}^*} - \epsilon > \max_{\pi:D_r(\pi,\pi^\dagger)<L^\pi} \mathcal{V}^\pi_{\mathcal{M}} - \Delta \cdot (L^\pi - D_r(\pi, \pi^\dagger))$, then the adversarial optimal policy is the policy with the highest policy value among the policies far from $\pi^\dagger$: $\hat{\pi}^* = \arg\max_{\pi:D_r(\pi,\pi^\dagger)=L^\pi} \mathcal{V}^\pi_{\mathcal{M}}$. Furthermore, any policy that is in the neighborhood of $\pi^\dagger$ is not $\epsilon$-optimal, so any $\epsilon$-optimal policy $\pi$ must be far from $\pi^\dagger$, i.e. $D_r(\pi, \pi^\dagger) = 0$.

Let $G_1$ be the event where the agent learns an $\epsilon$-optimal policy in the adversarial environment. When $G_1$ is true, the true performance of the policy learned by the agent is at most $\max_{\pi:D_r(\pi,\pi^\dagger)=L^\pi} V^\pi_{\mathcal{M}}$. When $G_1$ is false, the performance is at most $V^*$. By the efficient learning assumption, $G$ is true with probability at least $1 - p$, so we have $V \leq (1 - p) \cdot \max_{\pi:D_r(\pi,\pi^\dagger)=L^\pi} V^\pi_{\mathcal{M}} + p \cdot V^*$.

By the definition of attack, we have $B = |\Delta|$.

Let $G_2$ be the event where the agent takes actions with a distance to adversarial optimal actions less than $\delta$ in average. The adversarial optimal action at state $s$ is $\hat{\pi}^*(s)$, and the attack will not perturb the reward for any action $d(a, \pi^\dagger(s)) > r$. When $G_2$ is true, the agent will not take actions that have distance $d(a, \hat{\pi}^*(s)) > r_0$ for more than $\frac{\delta}{r_0} \cdot T$ times. When $r_0 = \min_s\{d(\hat{\pi}^*(s), \pi^\dagger(s)) - r\}$, the attack will not apply corruption if an action satisfies $d(a, \hat{\pi}^*(s)) < r_0$, so the attacker needs to apply corruption for at most $\frac{\delta}{r_0} \cdot T$ many rounds. When $G_2$ is false, the attacker applies corruption for at most $T$ rounds. The probability of $G_2$ to be true is at least $1 - p$, so the value of $C$ is bound by $C \le (1-p) \cdot \frac{\delta}{r_0} \cdot T + p \cdot T$ $\qquad \square$

**Proof for Theorem 5.7**

*Proof.* The policy value of a policy $\pi$ under the adversarial environment constructed by the AI attack with $\Delta$, $r$ and $\pi^\dagger$ satisfies: $\mathcal{V}^\pi_{\widehat{\mathcal{M}}} = \mathcal{V}^\pi_{\mathcal{M}} - \Delta \cdot D_r(\pi, \pi^\dagger)$. Note that for any policy that is in the neighborhood of $\pi^\dagger$, i.e. $D_r(\pi, \pi^\dagger) = 0$, its policy value is the same in both true and adversarial environments. If the following is satisfied: $\max_{\pi : D_r(\pi, \pi^\dagger)=0} \mathcal{V}^\pi_{\mathcal{M}} - \epsilon > \max_{\pi : D_r(\pi, \pi^\dagger)>0} \mathcal{V}^\pi_{\mathcal{M}} - \Delta \cdot D_r(\pi, \pi^\dagger)$, then the adversarial optimal policy is the policy with the highest policy value among the policies in the neighborhood of $\pi^\dagger$: $\hat{\pi}^* = \arg\max_{\pi : D(\pi, \pi^\dagger)=0} \mathcal{V}^\pi_{\mathcal{M}}$. Furthermore, any policy that is not in the neighborhood of $\pi^\dagger$ is not $\epsilon$-optimal, so any $\epsilon$-optimal policy $\pi$ must be in the neighborhood of $\pi^\dagger$, i.e. $D_r(\pi, \pi^\dagger) = 0$.

Let $G_1$ be the event where the agent learns an $\epsilon$-optimal policy in the adversarial environment. When $G_1$ is true, the true performance of the policy learned by the agent is at most $\max_{\pi : D_r(\pi, \pi^\dagger)=0} V^\pi_{\mathcal{M}}$. When $G_1$ is false, the performance is at most $V^*$. By the efficient learning assumption, $G$ is true with probability at least $1 - p$, so we have $V \le (1-p) \cdot \max_{\pi : D_r(\pi, \pi^\dagger)=0} V^\pi_{\mathcal{M}} + p \cdot T$.

By the definition of attack, we have $B = |\Delta|$.

Let $G_2$ be the event where the agent takes actions with a distance to adversarial optimal actions less than $\delta$ in average. The adversarial optimal action at state $s$ is $\hat{\pi}^*(s)$, and the attack will not perturb the reward for any action $d(a, \pi^\dagger(s)) \le r$. When $G_2$ is true, the agent will not take actions that have distance $d(a, \hat{\pi}^*(s)) > r_0$ for more than $\frac{\delta}{r_0} \cdot T$. When $r_0 = \min_s\{r - d(\hat{\pi}^*, \pi^\dagger(s))\}$, the attack will not apply corruption if an action satisfies $d(a, \hat{\pi}^*(s)) \le r_0$, so the attacker needs to apply corruption for at most $\frac{\delta}{r_0} \cdot T$ many rounds. When $G_2$ is false, the attacker applies corruption for at most $T$ rounds. The probability of $G_2$ to be true is at least $1 - p$, so the value of $C$ is bound by $C \le (1-p) \cdot \frac{\delta}{r_0} \cdot T + p \cdot T$ $\qquad \square$

## C Comparison to Reward Flipping Attack

Here we analyze and empirically examine the performance of a heuristic attack that appears to be effective as believed in Zhang et al. (2021b). The strategy of the attack is to change the sign of the true reward at each time. We call such attack as reward flipping attack, and its attack strategy can be formally written as $A^t(s^t, a^t) = -2r^t$. Note that such attack also construct a stationary adversarial reward function, suggesting that it also falls in our "adversary MDP attack" framework. Under such adversarial MDP $\widehat{\mathcal{M}}$, since all rewards have their signs flipped, we have $\mathcal{V}^\pi_{\widehat{\mathcal{M}}} = -\mathcal{V}^\pi_{\mathcal{M}}$ for all policy $\pi$, suggesting that the optimal policy under $\widehat{\mathcal{M}}$ actually has the worst performance under the true environment $\mathcal{M}$. However, the disadvantage of the attack method is that it breaks our second efficient attack principle where the perturbation for the adversarial optimal actions should be limited. The consequence is that it needs to apply corruption at every timestep, resulting in too much requirement on $C$. We further empirically test the efficiency of the reward flipping attack. The attacker cannot apply corruption when it runs out of budget on total corruption steps $C$, and we do not assume any constraint on $B$. We consider environment Hopper and HalfCheetah and set $C = 0.01$ as considered in our experiments while the remaining parameters are unchanged. Our results show that the performance of the reward flipping attack is comparable to the baseline UR attack as shown in the table 4 below, suggesting that the reward flipping attack is not efficient.

Table 4: Performance of reward flipping attack. The values in the table are the performance of the best policy ever learned by the learning algorithm, which is the same as the y axis of our main experiment results in figure 1

| Environment-Learning algorithm | Reward flipping attack | UR attack | No attack |
|---|---|---|---|
| Hopper-td3 | 3157 | 3482 | 3502 |
| Hopper-sac | 3521 | 3355 | 3496 |
| HalfCheetah-ddpg | 7463 | 6622 | 9341 |
| HalfCheetah-td3 | 7603 | 7694 | 9610 |

## D Comparison to VA2C-P attack

Here we compare the effectiveness of our AE attack and the most "black box" version of VA2C-P attack proposed in Sun et al. (2020) which knows the learning algorithm but does not know the parameters in its model. Note that the constraints for the two attacks are different in the two papers. To make sure that we are not underestimating the effectiveness of the VA2C-P attack, we let both attacks work under the same constraints used in Sun et al. (2020). The constraints here are characterized by two parameters $K$ and $\epsilon$. The training process is separated into $K$ batches of steps, and the attacker is allowed to corrupt no more than $C$ out of $K$ training batches. In each training batch with $t$ time steps, let $\delta\mathbf{r} = \{\delta r_1, \ldots, \delta r_t\}$ be the injected reward corruption at each time step, then the corruption should satisfy $\frac{\|\delta\mathbf{r}\|_2}{\sqrt{t}} \le \epsilon$. We modify the AE attack accordingly to work with such constraints. More specifically, the attack strategy is unchanged when applying corruption will not break the constraints, and we forbid the attack to apply corruption if doing so will break the constraints. To avoid anything that could cause a decrease in efficiency of VA2C-P attack, we do not modify any code related to training and attacking provided by Sun et al. (2020), and implement our attack method in their code.

Since VA2C-P has more limitations than AE attack, we only consider the scenarios where both attack are applicable. As an example, we choose Swimmer as the environment and PPO as the learning algorithm. Here we use the metric in Sun et al. (2020) to measure the performance of the attack. More specifically, we measure the average reward per episode collected by the learning agent through the whole training process. We set $K = 1, \epsilon = 1$, and the length of training to be 600 episodes where each episode consists of 1000 steps. The results for different attacks are shown in table 5. Each result is the average of 10 repeated experiment, and it is clear that our attack is much more efficient.

Table 5: Comparison between VA2C-P and AE attack

| clean | VA2C-P | AE | AE-(2) | AE-(3) |
|---|---|---|---|---|
| 30.07 | 25.49 | 14.75 | 7.08 | -2.16 |

We also notice that our AE attack computes faster than VA2C-P attack. To compute the attack for 600 training episodes in the experiment, our AE attack takes 135.7 seconds, while the VA2C-P black box attack takes 7049.5 seconds. In this case, the AE attack computes 52 times faster than VA2C-P attack.

One may notice that learning efficiency of PPO algorithm here is not as good as what we show in Figure 1. This is due to different implementation of the same algorithm in our work and Sun et al. (2020). It is a known issue that difference in implementations can lead to very different learning results Henderson et al. (2018). As mentioned before, we build our learning algorithms based on spinning up documents Achiam (2018), and the learning performance of our learning algorithms match the results shown in the spinning up documents.

# E    Framework of Online Reinforcement Learning under Reward Poisoning Attack

Here we present a detailed framework of the training process of a learning agent interacting with an environment $\mathcal{M} = (\mathcal{S}, \mathcal{A}, \mathcal{P}, \mathcal{R}, \mu_0)$ under reward poisoning attack considered in our work.

---

**Inputs** : Environment $\mathcal{M} = (\mathcal{S}, \mathcal{A}, \mathcal{P}, \mathcal{R}, \mu_0)$, Training steps $T$

**Initialize:** t=0 ;              /* The training starts at time step $t = 0$ */

**while** $t \leq T$ **do**

 $s^t \sim \mu_0$ ;         /* A training episode is initialized at state $s^t$ */

 $E = \text{False}$ ;  /* Environment termination signal has to be False at the beginning of
 each episode */

 **while** $E == \textit{False} \wedge t \leq T$ **do**

  Agent draws an action $a^t$ ;  /* Agent can run any reinforcement learning algorithm,
  such as PPO, to determine the action to draw. */

  $r^t \sim \mathcal{R}(s^t, a^t)$, $s^{t+1} \sim \mathcal{P}(s^t, a^t)$ ;   /* Environment generates the true feedback for
  instant reward and next state */

  Adversary determines reward perturbation $\Delta^t$

  Agent observes $(s^t, a^t, r^t + \Delta^t, s^{t+1})$

  **if** *Termination condition is reached* **then**

   $E = \text{True}$ ;  /* Environment checks if the termination condition is reached for
   this episode. The termination condition can be whether the maximum training
   steps in an episode is reached, or the agent enters a state $s^{t+1}$ that satifies
   certain conditions. */

  **end**

 **end**

**end**

---

Our AE and AI attacks define the strategy to determine the reward perturbation $\Delta^t$ in Definition 5.3, 5.6.

# F    Additional Discussion

**Potential defense for DRL against reward poisoning attacks:** Our work as well as previous RL attack works Rakhsha et al. (2020) show that a vulnerable learning algorithm fails to explore the actions of high interest enough times, resulting in low learning performance. So correspondingly, one idea for building robust algorithms is to make the exploration robust to the attack. For example, for the exploration idea following optimality in the face of uncertainty principle (e.g, UCB algorithms),the corruption of data can contribute additional uncertainty to the estimation of each action. By taking the additional uncertainty into account, the exploration strategy can explore the state and actions of high interest for enough times even if there can be adversarial attacks. As a result, the learning algorithm can be robust against attack. This idea has been applied in building robust learning algorithms in tabular MDP settings Wu et al. (2021). One can adopt a similar idea for building robust DRL algorithms in the future.

**How to reduce the requirement on $B$:** One can reduce the budget on B by using more budget on C. For example, one can follow the same idea of the AI attack to make the agent believe some random target actions to be optimal. In addition to decreasing the reward for non-target actions, one can also increase the reward for target actions. This can make the target policy even more appealing to the agent, so it requires less value of $B$ to influence the performance of a policy. Correspondingly, this will also significantly increase the requirement on $C$ as the attack needs to corrupt more steps.

