# OpenReview forum: "Efficient Reward Poisoning Attacks on Online Deep Reinforcement Learning"
_TMLR — Accepted by TMLR_

### Review · Reviewer_eEao · 2023-04-25

**Summary Of Contributions:**

The paper proposes a new reward poisoning attack against online RL. It proposes a new attack framework under certain assumptions. Under this framework, it proposes two attacks. The effectiveness of the proposed attacks are justified through both theoretical analysis and empirical evaluation.


**Audience:**

Yes

**Broader Impact Concerns:**

Not applied.

**Claims And Evidence:**

Yes

**Requested Changes:**

I would suggest the authors discuss some potential counter measures against their attacks.

**Strengths And Weaknesses:**

Strengths:
1. The paper proposes a new attack framework with two attacks. The techniques are solid with certain theoretical analyses.
2. The paper has conducted an extensive empirical evaluation on different RL environments.
3. The paper is overall well structured and written.

Weaknesses:
1. The reward poisoning attack against RL has been extensively studied in the existing literature
2. The paper lacks a discussion or evaluation of potential defense

---

> ### Author Response · Authors · 2023-05-29
>
> We thank the reviewer for their constructive comments!
>
> Q1:The reward poisoning attack against RL has been extensively studied in the existing literature
>
> A1:It is true that reward poisoning attack against RL has been well studied in the tabular RL settings. However, in the deep MDP setting, it is not well-studied in the literature. As we point out in Section 2, the attack methods for the tabular RL setting are not applicable to the DRL setting. As mentioned in Section 1, there are specific challenges for the attack in the DRL setting which are handled in this work.
>
> Q2:The paper lacks a discussion or evaluation of potential defense
>
> A2: It is true that the defenses against data poisoning attacks have also been well studied in simpler tabular MDP and bandits settings, but not in the Deep MDP setting. It has been shown that under the attack, a learning algorithm fails to explore the actions of high interest for enough times, resulting in low learning performance [1,2]. So correspondingly, one idea for building robust algorithms is to make the exploration robust to the attack. For example, for the exploration idea following optimality in the face of uncertainty principle (e.g, UCB algorithms), the corruption of data can contribute additional uncertainty to the estimation of each action. By taking the additional uncertainty into account, the exploration strategy can explore the state and actions of high interest for enough times even if there can be adversarial attacks. As a result, the learning algorithm can be robust against attack [3]. This idea has been applied in building robust learning algorithms in tabular MDP settings. One can adopt a similar idea for building robust DRL algorithms.
>
> [1]: Jun, K.S., Li, L., Ma, Y. and Zhu, J., 2018. Adversarial attacks on stochastic bandits. Advances in Neural Information Processing Systems, 31.
> [2]: Rakhsha, A., Radanovic, G., Devidze, R., Zhu, X. and Singla, A., 2020, November. Policy teaching via environment poisoning: Training-time adversarial attacks against reinforcement learning. In International Conference on Machine Learning (pp. 7974-7984). PMLR
> [3]: Wu, T., Yang, Y., Du, S. and Wang, L., 2021, July. On reinforcement learning with adversarial corruption and its application to block mdp. In International Conference on Machine Learning (pp. 11296-11306). PMLR

---

### Review · Reviewer_bYVw · 2023-05-06

**Summary Of Contributions:**

This paper studied reward poisoning attacks in reinforcement learning. The authors considered the black-box attack setting, where the attacker has no information about the learner and underlying environment. Two attack strategies are proposed - Action Evasion (AE) and Action Inducing (AI) attack. The AE attack tries to mislead the learning agent to believe that the optimal action in the clean environment is not optimal, thus the agent will be refrained from selecting the optimal action under attack. In contrast, the AI attack encourages the learning agent to behave like a random policy, taking random actions in every state, which results in a low-performance learned policy. The paper performed empirical study of AE and AI attack on diverse RL environments, and showed that both attacks can significantly reduce the performance of the learned policy. The authors also showed that AE and AI outperform the baseline random attack strategy.

**Audience:**

Yes

**Broader Impact Concerns:**

The paper does not have any ethical concerns.

**Claims And Evidence:**

Yes

**Requested Changes:**

(1). Please fix some definitions and notation issues. Please formally introduce the threat model and introduce symbols before using them,

(2). Consider improving the current theoretical results to give a better characterization on the performance drop induced by the AE and AI attack.

(3). It seems like reducing the per-step reward poisoning budget B causes the attack performance to drop quickly. How can we make the attack more effective subject to small B? Please provide some discussions on that.

**Strengths And Weaknesses:**

Strength:

(1). The attacker considered in this paper has more limited ability compared to prior works. First, the attacker does not know the underlying environment and the algorithm adopted by the learning agent. In this regard, a lot of prior works assume the attacker has access to a simulator of the underlying environment and the learner's algorithm, which allows the attacker to train a good poisoning strategy beforehand and then deploy the strategy in the true learning stage of the RL agent. However, this paper designs attacks that can be used on the fly during the training of an RL agent. Therefore, the attacks are more practical. Second, the paper considered budgeted reward poisoning both in terms of per-step reward perturbation and the attack frequency, which again makes the attack more stealthy in practice. Overall, the constraints on the attacker's ability makes the problem setup closer to a real world attack compared to existing methods.

(2). The paper provided some theoretical analysis on the performance of the attack, although the theoretical results do not look super intriguing to me (as explained later). These theoretical results provided some insights on the attack efficiency.

(3). The paper performed extensive empirical evaluations of the proposed AE and AI attacks. First, we see that the AE and AI attack are indeed very efficient. They require perturbation in less than 20% of training steps, but can significantly reduce the expected cumulative rewards of the learned policy. Second, we see that both AE and AI outperform the baseline random attack strategy. Finally, the paper also provided empirical study on different hyperparameters of attack, e.g., the per-step reward poisoning budget. The results look interesting and convincing.

Weaknesses:

(1). Some definitions are not clear enough. To list a few

The authors never specified clearly how the attack is performed in each time step. Assume the original reward is r_t. How does the attacker change the reward? Is it r_t+\Delta_t or r_t-\Delta_t? The authors seem to have used the symbol \Delta_t for reward perturbation without introducing it beforehand.

In Definition 4.1, it's not super clear what the expectation is with respect to. I suppose the expectation is with respect to the randomness in both attack strategy and the learning process. However, this is hidden from the definition.

The authors argue that Assumption 4.3 is reasonable. However, I think there are technical flaws in this assumption. The learned policy pi_0 is often random due to that the stochasticity in state transition and reward. Therefore, we cannot firmly claim that an agent can always learn an eps-optimal policy. In other words, the assumption 4.3 only holds probabilistically and is not strictly a reasonable assumption. At least the authors need to consider restating the assumption as some high-P event.

(2) While the paper derived interesting theoretical results to analyze the performance of the learned policy under attack, it seems really hard to see how much performance degradation is actually achieved under AE and AI attack. For example, in Theorem 5.5, the authors give an upper bound on V. However, it's not super clear that how far V is from original optimal value V*. Just from the theorem, it seems like V could be equal to V*, in which case the theory does not provide any guarantee in terms of the performance reduction after attack. This makes the theoretical results not super intriguing to me.

(3) In the experiments, the per-step reward poisoning budget seems too high to me. It can be as large as the maximum reward gap over all possible actions. This is almost like the attacker can arbitrarily change reward within the reward space.

---

> ### Author Response · Authors · 2023-05-29
>
> We thank the reviewer for their constructive comments!
>
> Q1:The authors never specified clearly how the attack is performed in each time step.
>
> A1: We formally introduce the attack model in Section 3 in the revision. At training step $t$, let the state and action taken by the agent be $s^t$ and $a^t$. The environment generates an instant reward $r^t$ and next state $s^{t+1}$. The adversary observes $(s^t,a^t,r^t,s^{t+1})$, and injects a perturbation $\Delta^t$ to the reward signal, then the agent observes perturbed observation $(s^t,a^t,r^t+\Delta^t,s^{t+1})$.
>
> Q2:In Definition 4.1, it's not super clear what the expectation is with respect to.
>
> A2:It is correct that the expectation is with respect to the learning process in the environment under the attack. We clarify this in the revision for Definition 4.1.
>
> Q3:The authors argue that Assumption 4.3 is reasonable. However, I think there are technical flaws in this assumption.
>
> A3: We modify assumption 4.3 by making the guarantee a high-probability event. Accordingly, our main theoretical results on the attacks are slightly modified. In short, when the agent fails to learn well in the adversarial environment with probability $p \ll 1$, in the worst case, it may happen to learn the actual optimal policy with performance $V^*$, and the attack applies corruption at all $T$ step. The bound on $V$ and $C$ is then modified to include this low-probability case. Also, the definition of the trivial attack is modified. Under the modified assumption, when there is no attack, the agent may learn the worst policy with a probability less than $p$. The detailed changes in our assumptions, definitions, and theorems can be found in the revision in sections 4 and 5. The proofs are also changed accordingly and can be found in Appendix B in the revision.
>
> Q4:While the paper derived interesting theoretical results to analyze the performance of the learned policy under attack, it seems really hard to see how much performance degradation is actually achieved under AE and AI attacks.
>
> A4: For the AE attack, the bound we give on $V$ in the theorem is $V \leq (1-p) \cdot \mathcal{V}^{\hat{\pi}^*}_{\mathcal{M}} + p \cdot V^*$, and $\hat{\pi}^*$ is the policy of the highest performance among a specific set of policies $\pi:D_r(\pi,\pi^\dagger)>0$. These policies are restricted to only taking actions that are far from some target actions. Our theorem implies that with a high probability, the learning agent will learn a policy from one of these policies. Similarly, for the AI attack, the bound of $V$ is given by the highest performance from the set of policies which only take action close to some target actions. These sets of policies are very restricted compared to the whole policy space. Since we have no information on the environment dynamics, it is impossible for us to give any further theoretical analysis of the specific policy sets. However, we argue that intuitively, the performances of the policies of the two sets are usually low in practice. Recall the policy set under the AI attack consists of policies similar to a random policy. For most practical DRL environments, such policies usually have low performance. Even for the simplest CartPole problem among the classical control problems from gym, taking random actions for every step almost always lead to low total rewards in the end. The policy set under the AE attack consists of policies that never take some actions. Every time when the optimal action is not taken, the long-term reward for the policy is decreased. This accumulates over each step, resulting in a low total reward in the end. For example, in the MountainCar problem, the key to achieving high performance is to accumulate the energy of the car as fast as possible. At each state, only one action can increase the energy, and the others cannot change or even decrease the energy. If the car fails to gain energy at many states, it can be much slower for it to gain enough energy for reaching the top, resulting in a low performance.

---

> ### Author Response · Authors · 2023-05-29
>
> Q5:In the experiments, the per-step reward poisoning budget seems too high to me. It can be as large as the maximum reward gap over all possible actions.
>
> A5: In the simpler tabular MDPs, it has been shown that an attacker requires a relatively high budget on per-step corruption $B$ to make the agent learn an arbitrary target policy efficiently [1]. Even with full knowledge of the environment dynamics and the learning algorithm, the attack there requires $B$ as large as the maximum reward gap over all possible actions. Moreover, there is an additional difficulty in our setting compared to the tabular settings considered in previous works [1,2]. We consider the discounted infinite MDP setting. For some of the environments considered in our work, there exist some states called absorbing states. These are the states that satisfy the termination condition of the environment. Once an agent reaches such an absorbing state, it will always remain in the state and receive $0$ rewards regardless of the actions it takes afterward. In practice, the training episode will restart from an initial state when the agent reaches an absorbing state. Therefore, our attack is not able to corrupt the steps after an agent reaches an absorbing state. As a result, while the performance of a policy is determined by the discounted reward from infinitely many steps, our attack is only able to influence it on a limited number of steps, so the attack in our setting requires a high budget of $B$ to influence the performance of a policy.
>
> Q6:It seems like reducing the per-step reward poisoning budget B causes the attack performance to drop quickly. How can we make the attack more effective subject to small B?
>
> A6:One can reduce the budget on B by using more budget on C. For example, one can follow the same idea of the AI attack to make the agent believe some random target actions to be optimal. In addition to decreasing the reward for non-target actions, one can also increase the reward for target actions. This can make the target policy even more appealing to the agent, so it requires less value of $B$ to influence the performance of a policy. Correspondingly, this will also significantly increase the requirement on $C$ as the attack needs to corrupt more steps.
>
> [1]: Zhang, X., Ma, Y., Singla, A. and Zhu, X., 2020, November. Adaptive reward-poisoning attacks against reinforcement learning. In International Conference on Machine Learning (pp. 11225-11234). PMLR.
> [2]: Rakhsha, A., Radanovic, G., Devidze, R., Zhu, X. and Singla, A., 2020, November. Policy teaching via environment poisoning: Training-time adversarial attacks against reinforcement learning. In International Conference on Machine Learning (pp. 7974-7984). PMLR

---

### Review · Reviewer_bozW · 2023-05-16

**Summary Of Contributions:**

The authors introduce a black box framework for reward poisoning attacks in training time. Within this framework, they develop two specific attacks that manipulate the reward signals in deep RL during a small portion of the training process, leading the agent to learn a suboptimal policy. The authors provide a theoretical analysis to assess the effectiveness of their attack strategy and conduct extensive empirical evaluations. The results obtained indicate that their attacks can effectively undermine the learning capabilities of various popular DRL algorithms.

**Audience:**

Yes

**Broader Impact Concerns:**

I don't have concerns about the ethical implications of this work.

**Claims And Evidence:**

Yes

**Requested Changes:**

Please see the Weakness section above. Specifically, I think more explanation on the values of $\epsilon$ and $\delta$ in the assumption, and pseudocode for the algorithms are welcomed.

**Strengths And Weaknesses:**

Strength:
> I found this work well-written and easy to follow. Reward poisoning attack is not well-studied in the literature. The authors first use the uniformly random time attack as a baseline and show its weakness, then they list out the principles of a non-trivial attack.
> Two main attack frameworks are introduced intuitively. I checked the proofs in the appendix and they seem to be correct.
> The authors conducted extensive experiments to show the efficacy of the proposed methods.


Weakness:
> I found Assumption 4.3 a bit too strong, especially the claim that both $\epsilon$ and $\delta$ <<1. Do the authors have any intuitions or examples for the numerical values of $\epsilon$ and $\delta$? Also, the notation of $\epsilon$ in this assumption overlaps with the $\epsilon$-greedy, so I think it's better to use another Greek letter here.
> Both AE and AI attacks are described in definitions, which may cause difficulties in understanding. I think it would be better to have a pseudo-code for how they are implemented in a typical DRL algorithm, for example, PPO. Also, for the theorem 5.5 and 5.7, it seems that $\min_s\{ r−d(π^∗(s),π(s)) \}$ can be a very small number, and thus the upper bound can be very loose.
>
> Minor: please refrain from only using color to distinguish curves in Figures 3 and 6, as it is not friendly to readers with color blindness.

---

> ### Author Response · Authors · 2023-05-29
>
> We thank the reviewer for their constructive comments!
>
> Q1:I found Assumption 4.3 a bit too strong, especially the claim that both $\epsilon$ and $\delta \ll 1$. Do the authors have any intuitions or examples for the numerical values of $\epsilon$ and $\delta$?
>
> A1: As mentioned by reviewer bYVw, we make assumption 4.3 more realistic by making the guarantee from the learning algorithm a high probability event with probability at least $1-p$ and $p \ll 1$. The details can be found in section 4 of the revision. Note that this leads to some small modifications in the theorems, and the details can be found in Section 5 in the revision. The value of $\epsilon$ represents how close the performance of the learned policy can be to that of the optimal policy with a high probability $1-p$. $\epsilon$ should be small as the goal of a learning algorithm is to learn a good policy with a high chance. The value of $\delta$ represents how close the actions taken by a learning algorithm are to the optimal actions with a high probability $1-p$. To handle the exploration-exploitation trade-off, the exploration strategies tend to take actions close to the optimal actions more often. For example, for $\epsilon$-greedy exploration that is commonly used in practice, the agent will only take an empirically non-optimal action with a small chance at every time. If the agent is assumed to learn the optimal actions successfully during training with a high chance, then the value of $\delta$ is likely to be small. In the simpler tabular MDP setting, the efficient learning algorithms have a low-regret guarantee [1] which implies a low value of $\epsilon$ and $p$, and for the environments where different actions lead to results that are not similar, the value of $\delta$ will also be small [2]. Due to a lack of theoretical understanding of DRL algorithms, there is no theoretical guarantee on the values of $\epsilon$, $p$ and $\delta$. The details can be found in Section 4.
>
> Q2: Also, the notation of $\epsilon$ in this assumption overlaps with the $\epsilon$-greedy, so I think it's better to use another Greek letter here.
>
> A2: Since we only mention $\epsilon$-greedy once in our work for explaining the intuition behind assumption 4.3, we use $E$-greedy (uppercase of $\epsilon$) instead to avoid confusion in the revision.
>
> Q3:Both AE and AI attacks are described in definitions, which may cause difficulties in understanding. I think it would be better to have a pseudo-code for how they are implemented in a typical DRL algorithm,
>
> A3: We formally introduce the threat model in Section 3 in the revision so one can see clearly how corruption is injected during training.  Note that the attack is applied to the environment dynamics instead of the learning algorithm directly. So we show a detailed framework of how the attack influences the training of an agent in an environment with an arbitrary learning algorithm in Appendix E in the revision.
>
> Q4:Also, for the theorem 5.5 and 5.7, it seems that $\min_s r-d(\pi^*(s),\pi^\dagger(s))$ can be a very small number, and thus the upper bound can be very loose.
>
> A4:In the discrete action space case, for the AI attack, we have $d(\pi^*(s),\pi^\dagger(s))=0$ for any state. Since any value of $r \in (0,1)$ leads to the same AI attack, we can take $\min_s r-d(\pi^*(s),\pi^\dagger(s)) = 1$ in the bound formulation to achieve a tighter bound $C \leq (1-p) \cdot \delta \cdot T + p \cdot T$. Similarly for the AE attack, in the discrete action space case, we can also take $\min_s d(\pi^*(s),\pi^\dagger(s)) - r = 1$ in the bound formulation. We make this explicit in the revision in section 5. For the continuous action space case, the actual bound depends on the state distribution determined by the interaction between the learning algorithm and the environment. Since we have no knowledge of the environment dynamics and specific knowledge of the learning algorithm, it is impossible to evaluate the state distribution. So we approximate the bound by only considering the worst case where the learning algorithm always visits the state $\text{argmin}_s r-d(\hat{\pi}^*(s),\pi^\dagger(s))$. It is possible that with more knowledge of the environment dynamics or the learning algorithm, one can find a tighter bound on it in the continuous action space case.
>
> Q5: Minor: please refrain from only using color to distinguish curves in Figures 3 and 6, as it is not friendly to readers with color blindness.
> A5: We will use dashed and solis lines for the curves in Figure 3 and 6 in the next version.
>
> [1]: Jin, C., Allen-Zhu, Z., Bubeck, S. and Jordan, M.I., 2018. Is Q-learning provably efficient?. Advances in neural information processing systems, 31
>
> [2]: Rakhsha, A., Radanovic, G., Devidze, R., Zhu, X. and Singla, A., 2020, November. Policy teaching via environment poisoning: Training-time adversarial attacks against reinforcement learning. In International Conference on Machine Learning (pp. 7974-7984). PMLR

---

### Author Response · Authors · 2023-05-29

Thanks to all reviewers for their constructive comments! We make some changes in the paper based on the comments, which are highlighted in red in the revision.

---

### Author Response · Authors · 2023-06-13

Dear Reviewers,

Thanks again for your constructive reviews! Please let us know if you have any follow-up comments or concerns for our responses and revision. We are very happy to answer any further questions.

---

### Decision · Action_Editors · 2023-06-25

**Recommendation:** Accept with minor revision

**Comment:**

The paper proposes two reward-poisoning attacks strategies in a general RL setting. Reviewers agree the work has novel and substantial contributions, with both theoretical and empirical supports. The revised version satisfactorily addresses major questions raised in the initial reviews, such as assumption 4.3 and a few places for greater clarity. The AE agrees with the reviewers, and has a few more suggestions:

1. The authors promised to change the line types in Figures 3 and 6. But the revised version seems to have the same plots. Can you fix them?

2. The paper highlights "deep" RL throughout. I find it puzzling, because the only connection between deep RL and this work is that the authors consider non-tabular RL settings. If so, why not just call it general RL, or RL with function approximation? Calling non-tabular settings as "deep" is a misnomer IMHO.

3. (Minor) Does it make (mathematical) sense to change equality constraints to inequality ones in (2), as well as other parts?

**Audience:**

Reward poisoning attacks in RL is receiving growing interests, and this paper gives a solid study in a setting that is more general than previous work. It's expected to be interesting to a reasonably large audience in the AI community.



**Claims And Evidence:**

The paper proposes two attack strategies (AE, AI) in a general RL setting. It provides theoretical and empirical evidence to show the attacks are effective at lowering the learned policy's performance.

---

> ### Author Response · Authors · 2023-07-13
>
> We thank the action editor and the reviewers for providing constructive feedback that has helped improve our paper. We have included the latest suggestions in the camera ready version and submitted it. We summarize these changes below:
>
> 1. We have changed the line styles in the camera-ready version.
> 2. We highlight ‘deep’ because the focus of our work is on the reinforcement learning problems that require learners to use deep neural networks for function approximation. The main results in the experiments are all about DRL algorithms learning in complicated environments that are hard to solve without using deep neural networks for function approximation. The main difficulties and challenges in our problem also arise from the use of deep networks. The background and the formulation of our problem are, however, general RL settings, so we modify it accordingly and make it clear in the camera-ready version.
> 3. We also considered using inequalities in Eq (1),(2) in the beginning but decided to use equalities eventually because equalities can better reflect how exactly efficient an attack method is. With inequalities, two attacks with different strengths can satisfy Eq(2) with the same coefficients $(V,B,C)$.